# Negative Ion Purifier Effects on Indoor Particulate Dosage to Small Airways

**DOI:** 10.3390/ijerph19010264

**Published:** 2021-12-27

**Authors:** Mengjie Duan, Lijuan Wang, Xingyan Meng, Linzhi Fu, Yi Wang, Wannian Liang, Li Liu

**Affiliations:** 1Vanke School of Public Health, Tsinghua University, Beijing 100084, China; mengjie_archi@tsinghua.edu.cn; 2Department of Building Science, Tsinghua University, Beijing 100084, China; 3State Key Laboratory of Green Building in Western China, Xi’an University of Architecture and Technology, Xi’an 710055, China; wanglijuan@live.xauat.edu.cn (L.W.); mxy@live.xauat.edu.cn (X.M.); fulinzhi@xauat.edu.cn (L.F.); wangyi@xauat.edu.cn (Y.W.); 4School of Building Services Science and Engineering, Xi’an University of Architecture and Technology, Xi’an 710055, China; 5Laboratory of Eco-Planning & Green Building, Ministry of Education, Tsinghua University, Beijing 100084, China

**Keywords:** indoor air quality, airway replica, inhalation exposure, size distribution

## Abstract

Indoor air quality is an important health factor as we spend more than 80% of our time indoors. The primary type of indoor pollutant is particulate matter, high levels of which increase respiratory disease risk. Therefore, air purifiers are a common choice for addressing indoor air pollution. Compared with traditional filtration purifiers, negative ion air purifiers (NIAPs) have gained popularity due to their energy efficiency and lack of noise. Although some studies have shown that negative ions may offset the cardiorespiratory benefits of air purifiers, the underlying mechanism is still unclear. In this study, we conducted a full-scale experiment using an in vitro airway model connected to a breathing simulator to mimic inhalation. The model was constructed using computed tomography scans of human airways and 3D-printing technology. We then quantified the effects of NIAPs on the administered dose of 0.5–2.5 μm particles in the small airway. Compared with the filtration purifier, the NIAP had a better dilution effect after a 1-h exposure and the cumulative administered dose to the small airway was reduced by 20%. In addition, increasing the negative ion concentration helped reduce the small airway exposure risk. NIAPs were found to be an energy-efficient air purification intervention that can effectively reduce the small airway particle exposure when a sufficient negative ion concentration is maintained.

## 1. Introduction

Many cities in China are continuously affected by smog, and the air is heavily polluted. As a result, the prevalence of respiratory diseases has been drastically increasing. A World Health Organization report has shown that respiratory infections cause approximately four million deaths annually, which account for 7% of deaths worldwide [1]. We spend more than 80% of our time indoors and indoor particulate matter (PM), particularly fine particle matter (PM_2.5_), has been causally linked to respiratory disease [2,3,4]. Indoor PM is a complex mixture with varied compositions, origins and adverse health effects; the efficient control of indoor particulate exposure can substantially improve public health [5].

Indoor air purifiers are the most common intervention strategy adopted to reduce indoor PM_2.5_ exposure, with filtration purifiers being the traditional type of indoor air purifier used [6,7]. Intervention studies have reported the cardiopulmonary benefits of high-efficiency particulate air (HEPA) filtration purifiers in young, healthy adults in a Chinese city with severe ambient particulate air pollution [8]. However, in recent years, negative ion air purifiers (NIAPs), which boast the advantages of low energy consumption, no noise and convenient installation, have become increasingly common in residences, offices, schools and other indoor settings [9].

Negative ions released by NIAPs enhance the coagulation of inhalable airborne particles. Due to their larger size and consequently larger terminal velocities, coagulated particles can be more easily sedimented and removed from the air. Previous studies have confirmed the purification performance of NIAPs, which can more efficiently remove PM_2.5_ from the air than traditional filtration purification systems [9,10,11,12,13]. Numerous studies have discussed the health consequences of exposure to airborne ions and reported conflicting conclusions. Some studies have found that exposure to negative air ions has a beneficial effect on respiratory health [14,15,16]. In contrast, other studies have reported that exposure to ions adversely affects lung function and cardiovascular performance, causes irritation and potentially exacerbates asthma symptoms [16,17,18,19,20]. Unfortunately, due to limited sample sizes and/or significant methodology biases in these studies, the health effects of NIAPs remain unclear and controversial [21].

Recent advantages of low energy consumption and high purification performance have renewed public interest in NIAPs. A recent systematic review suggested that high airborne ion exposure is associated with lower depression symptoms but not with anxiety, mood, relaxation, sleep or personal comfort measures [22]. Furthermore, high airborne ion exposure is not associated with any therapeutic benefits, changes in respiratory function or symptomatic outcomes [21]. In summary, some studies have reported pulmonary benefits after negative ion exposure; however, meta-analyses have not found reliable evidence supporting the effects of NIAPs on respiratory or metabolic outcomes. 

Epidemiological assessments have used the gold standard of randomised controlled trials instead of observational studies to evaluate the health outcomes of NIAPs. Wallner et al. [23] quantified the short-term effects of air ions on physiological and psychological parameters and observed slightly activating and cognitive performance-enhancing effects of higher indoor air ion concentrations. However, no influences of air ions on lung function and well-being were detected. In addition, 62 animal studies have been unable to detect a biological mechanism of interaction and no evident dose–response relationships between beneficial health effects and negative air ions have been reported [24].

Nevertheless, two recent NIAP intervention studies have yielded different results. Dong et al. [20] conducted a random double-blind crossover trial on a group of school children using NIAPs. NIAPs were found to significantly reduce the concentrations of indoor PM. However, high negative ion concentrations had negative impacts on heart rate. Liu et al. [19] performed a week-long intervention with NIAPs in the dormitories of 56 healthy college students. Each student received one real and one sham (NIAP not activated) intervention in their dormitory. Lung function and cardiovascular biomarkers were assessed before and after each intervention. The NIAP intervention contributed to high negative ion concentrations and low PM_2.5_ concentrations in the indoor environment. However, an association between negative ion concentration and increased systemic oxidative stress was also observed. Thus, high indoor negative ion concentrations may offset the beneficial effects associated with reduced PM_2.5_. 

A recent systematic review and meta-analysis comprehensively investigated the cardiovascular effects of reducing PM_2.5_ exposure through indoor air purification. Based on the meta-analysis of 14 independent randomised controlled trials, NIAPs appear to have short-term cardiovascular benefits [10]. However, the overall certainty of evidence remains low due to a range of study limitations [10]; more extensive and robust studies are needed to clarify the health outcomes of NIAPs.

Dose assessments can elucidate causal relationships between environmental exposure, inhalation doses, small airway doses and associated health outcomes. Due to the limitation of in vivo methods, the effects of NIAPS on PM_2.5_ human inhalation doses and small airway doses have not yet been quantified [25]. 

This study establishes an in vitro method for quantifying PM_2.5_ exposure doses after air purification interventions (NIAPs and filtration purifiers) in a full-scale cleanroom. A three dimensional (3D)-printed human replica, including the face, oropharynx, trachea, five bronchi generations and lung volumes, was developed to mimic natural human exposure scenarios. We measured the exposure dose in the breathing zone and the administered dose in the small airways within 1 h of inhalation and compared the doses between NIAPs and filtration purifiers with the same clean air delivery rate (CADR). In addition, we evaluated the effects of different negative ion concentrations on particle purification and human exposure. Our findings provide a specific dose reference for in-depth evaluations of NIAP effects on human respiratory health. 

## 2. Materials and Methods

### 2.1. Measuring the Administered Dose in the Small Airway

We developed a 3D-printed human replica (Figure 1) based on computed tomography (CT) scans of the small airways of a healthy 34-year-old male, who is a non-smoker with no airway disease, to quantify the administered PM_2.5_ dose [26]. The model includes the face, nasopharynx, trachea, G1–G5 bronchi and lung cavities. The surface roughness of the model is less than 0.1 mm and the electrostatic capacitance is less than 0.002 μC (< 0.4 m/s flow speed; 25 °C, 50% relative humidity). A breathing airflow simulator was connected with the model to provide periodic inhalation airflow. The airflow rate *Q* was 13.4 L/min, which mimicked a healthy standing male [27]. The inhalation frequency *f* was 15/min. We calculated the administered dose in the small airway as the accumulated mass of particles that penetrated the G5 bronchi to enter the lung cavities during the exposure time.

### 2.2. Experimental Set-Up

The full-scale experiment was carried out in an ISO-2 cleanroom (5 × 3.5 × 2.5 m). An air-conditioning system YSM50M-0713-S-L (YORK Incorporated, Qingdao, China; 380 V, 3 N, 50 Hz), with a rated ventilation rate of 4500 m^3^/h, a cooling capacity of 30 kW and a heating capacity of 18 kW, was used to control the indoor temperature and humidity in the room. The room temperature was adjusted to 24–26 °C and the relative humidity was set to 40–70% (summer condition, thermal comfort level I, −0.5 ≤ Predicted Mean Vote ≤ +0.5) [28]. The cleanroom was ventilated by mixing, with an air change rate of 2/h. Each outlet had a HEPA filter with a filtration efficiency of >99.5% for particles > 0.3 µm. There was no air recirculation during the measurements. 

We selected a commonly used filtration purifier AC4025 (PHILIPS Incorporated, Amsterdam, The Netherlands) and a NIAP Bentax® A6E (Varionix GmbH, Cham, Switzerland) for comparison. The Clean Air Delivery Rates (CADRs) of the filtration purifier and NIAP were 148 m^3^/h and 142 m^3^/h, respectively. The standard particulate source was a Hongtashan cigarette (tar amount = 8 mg) [29]. Two cigarettes were placed near the ventilation outlet and two five-blade fans (FG11-42D, 0.42 m diameter, 20 W) were placed above the particulate source. The particulate pollutants were evenly mixed in the room to a concentration of ~1000 pt/cm^3^.

The negative ion concentration C_NIAP_ was detected using an atmospheric negative oxygen ion device COM-3500C (Grows Instrument, Shanghai, China). The measurement range was 0–1 million/cm^3^ with an error of ±5%. We used an Aerodynamic Particle Sizer (APS) model 3321 (TSI Incorporated, Shoreview, MN, USA) to monitor the particulate size distribution and concentration in real-time. The APS provides count size distributions for particles with aerodynamic diameters ranging from 0.523 to 19.810 μm. The administered dose in the small airway of the in vitro model was measured by connecting the APS to the lung cavity sampling port. 

We measured the PM environmental exposure dose in the breathing zone and the administered dose in the small airway. We compared three interventions: no purifier, filtration purifier and NIAP. The in vitro model was placed 1.0 m away from the purifiers. In addition, we quantified the influence of negative ion concentrations on the administered dose by changing the distance between the NIAP and the in vitro model to achieve concentrations of 7.0 × 10^5^ pt/cm^3^, 3.0 × 10^5^ pt/cm^3^ and 1.5 × 10^5^ pt/cm^3^. Each condition was repeated three times. The experimental set-up is illustrated in Figure 2 and experimental parameters are listed in Table 1.

### 2.3. Particulate Matter Exposure Dose

The APS was used to measure the PM concentration in real-time and the particle dose in the breathing zone and small airway was calculated as the cumulative mass (Equation (1)).
(1)$$M=\overset{t=3600}{\underset{t=1}{\sum }}dN\left(t\right)\left(\frac{\pi }{6}\right){\left(D_{a}\sqrt{\frac{\rho_{0}}{\rho_{1}}}\right)}^{3}\rho_{0}q_{inh}T_{sample}$$
where *dN*(*t*) is the total number of particles per unit volume of sampled air (pt/cm^3^); *D_a_* is the particle aerodynamic diameter (μm); *ρ*_0_ is the particle density, 1.18 g/cm^3^ of cigarette smoke; *ρ*_1_ is the unit density, 1.0 g/cm^3^; *q_inh_* is the inhalation airflow rate, 13.4 L/min; *T_sample_* is the sampling time, 1 s.

### 2.4. Blank Measurement

We measured the PM_10_ concentration in the breathing zone in the no-purifier, filtration purifier and NAIP purifier conditions. The particle concentration decay within 1 h was recorded, as shown in Figure 3. The first-order exponential decay equation (Equation (2)) was used to obtain the best-fitting curve of the particle concentration. The constant decay (*k*), which represents the purification ability of purifiers, was then calculated.
(2)$$\;C_{N}=C_{0}\exp\left(-kt\right)$$
where *C**_N_* is the particle concentration at time *t*, pt/cm^3^; *C*_0_ is initial particle concentration, pt/cm^3^; *k* is the dimensionless decay constant; *t* is sampling time, min.

Figure 3 shows the concentration decay fitting curves for the three conditions (fitting constant R^2^ > 0.99). The particle concentration within the same exposure time in the breathing zone of either filtration purifier or NAIP was significantly lower than in the no-purifier condition. Thus, both the filtration purifier and NAIP exerted purification effects on environmental particulates. The constant decay *k* of NIAP was 0.079 and 1.23 times that of the filtration purifier, indicating a better purification ability for PM_10_.

## 3. Results

### 3.1. Inhalation Exposure

Without air purification, the potential inhalation dose of PM_2.5_ was 19.25  ±  1.87 µg (Figure 4). We expected particle concentrations to be rather uniform outside the regions adjacent to the particle sources because the particles were well-mixed by the ceiling-mounted fans. Once inhaled, particles were deposited on the inner surfaces of the airway replica. Particles that penetrated the partial bronchi then entered the lung cavities. 

We assumed that the particle concentrations in the joint conducting tube were equal to the average concentration in the lung cavity by neglecting the particle deposition on the inner surface of the lung boundaries, the outer surface of the partial bronchi and the inner surface of the conducting tubes attached to the breathing airflow simulator. The dose of particles in the small airway, where the bronchi have a diameter less than 1.5 mm, was 15.70  ±  1.07 µg, as estimated by Equation (1). The average deposition ratio of particles on the upper and central airway was 1–15.70/19.25 = 18.44%, which is consistent with previous studies based on casting models of the human airway [30]. 

The two air purification interventions had quite similar removal abilities with respect to the potential inhalation dose in the well-mixed condition in the test room. The filtration purifier’s higher CADR value (148 m^3^/h) resulted in a slightly higher potential inhalation dose (11.07 ± 0.26 µg) compared to the NIAP (10.07 ± 0.72 µg), which had a CADR value of 142 m^3^/h. The performance of the NIAP was slightly less consistent than that of the filtration purifier, which could be due to the potential variation in negative ion concentrations between repetitions. 

The inhaled particles were smaller after filtration, which reduced the deposition ratio in the upper airway, i.e., most inhaled particles reached the small airway. In contrast, NIAP reduced the dose delivered to the small airway by coagulating fine particles into coarse particles, as shown in Figure 5. NIAP effectively reduced the small airway exposure dose by 20.3% compared to the filtration purifier with a similar CADR.

### 3.2. Effect of Negative Ion Concentration on Coagulation Size

The coagulation effect of the NIAP was evaluated as the particle number ratio variation, i.e., how the size distribution varied based on the NIAP condition (on or off). In the presented setup, the number of particles below 0.626 µm decreased and the number of particles between 0.626 and 1.596 µm increased, revealing a shift in the size distribution towards larger particles when the NIAP was on. Notably, the variations in particle number ratio were less than 3% for all size bins; the variations in dose reached 47.69% in the breathing zone and 45.35% in the small airway (Figure 4). Thus, a small shift in size distribution via coagulation resulted in a significant reduction in exposure.

The relative number density between negative ions and particles determines how many particles in specific size ranges can coagulate into larger particles, as shown in Figure 6. We adjusted the distance between the NIAP and the subject to adjust the negative ion concentration in the breathing zone and airway. The effect of relocating the NIAP on the particle source was omitted due to the overwhelming levels of particle emission and fan mixing.

Low and high concentrations of negative ions affect particle adhesion, consequently shifting the particle size distributions. Particles larger than 0.835 µm were insensitive to the negative ion concentration change from 0 to 1.5 × 10^5^ pt/cm^3^ as the particle number ratio variation between NIAP being on and off was negligible. The critical size bin that differentiated the ratio decrease from the ratio increase revealed the effect of negative ions on particle coagulation. Increasing the negative ion concentration increased the larger particle coagulation, i.e., the critical size bin was larger. In the breathing zone, the critical size bin was 0.583 µm at a negative ion concentration of 1.5 × 10^5^ pt/cm^3^. The critical size bin increased to 0.626 µm when the negative ion concentration doubled but remained at 0.626 µm when the ion concentration was doubled again. In the airway, the critical size bin was 0.583 µm at negative ion concentrations of 1.5 × 10^5^ and 3.0 × 10^5^ pt/cm^3^ and increased to 0.626 µm at a negative ion concentration of 7.0 × 10^5^ pt/cm^3^. 

### 3.3. Effect of Negative Ion Concentration on Inhalation Exposure

Figure 7 illustrates the reverse linear correlation between the negative ion concentration and particle dosage. The potential inhalation dose was reduced by 12.7% and the small airway dose was reduced by 19.7% when the negative ion concentration increased from 1.5 × 10^5^ to 7.0 × 10^5^ pt/cm^3^. As shown in Figure 6, the shifts in size distribution by coagulation were below 5% in all size bins, but these changes resulted in significant reductions in exposure. 

## 4. Discussion

Many countries around the world are experiencing a heavy disease burden due to air pollution. Traditionally, air purification is expensive both economically and in terms of energy consumption. The latter factor is complicated as energy production and consumption also produce particle emissions. In our measurements, NIAPs required less than one-sixth of the energy consumed by filtration purifiers. This advantage is attractive as it offers a potential solution to meet health demands without creating an additional energy burden.

This study provides new evidence of particle dosages delivered to small airways with different air purification interventions. In this context, the NIAP outperforms the filtration purifier at similar CADRs, provided negative ion concentration is fixed. The effects of relocating the NIAP on the particle source may in turn affect the particle dosage, but we include this possibility in the discussion of negative ion concentrations, as shown in Figure 6. In Figure 6a, the variation in the particle number ratio has a bimodal distribution with the particle size. The first peak between 0.626 and 0.835 µm was probably formed due to the coagulation of particles around the breathing zone, whereas the second peak between 1.037 and 1.596 µm might have been formed by the coagulation of particles at the emission phase due to the proximity of the particle source to the NIAP. The protective effect of the NIAP was more pronounced when it was located closer to the subject than to the source. This observation requires further examination with the inclusion of additional factors such as the subject’s lung function, room size and ventilation rate. 

Recent research that assessed the potentially harmful effects of negative ions on the cardiovascular system by examining biomarkers has added new cause for caution in implementation. Although the present study demonstrated the benefit of reduced PM_2.5_ dosage using the NIAP, the overall health consequences may not be reflected in our model. The dose we measured was the administered dose and not the biological effective dose, which quantifies the particles entering the inner microenvironment of the airway and lung epithelium cells. The charge of the particle added by ions may also enhance the permeability of fine particles, thereby increasing the biological effective dose.

The in vitro airway replica adopted in this experiment was reconstructed from CT scans of the airways of a human subject using 3D-printing technology. Thus, the geometric boundary was similar to real airways and could restore periodic inhalation airflow and particle movement. Due to the limitation of CT-scan resolution, only the first five generations of the bronchial structure were reconstructed. The lack of the remaining bronchi introduced uncertainty in both airflow and particle movement in the fifth-generation bronchi as the subsequent boundary conditions were absent. The effect of this limitation is not significant as we specifically aimed to measure the total quantity of particles leaving the fifth-generation bronchi or the total quantity of particles arriving at sixth-generation bronchi, i.e., the small airway denoted in this study [31]. In addition to the missing bronchi, we did not reconstruct airway cilia or mucous membranes. This simplification should not result in significant error in terms of particle deposition as the boundary roughness of the replica was specifically improved to less than 1 µm. However, the lack of mucous affects the mucous clearance mechanisms that can relocate physical and biological dosages.

Our airway model simulated steady-state exposure without lung contraction and expansion and lacked the temperature and humidity environment inside the human body. Therefore, variations in the interception and inertial impaction caused by the narrowing of airway boundaries and condensation growth could not be mimicked by our airway reconstruction model. However, the deposition mechanisms of turbulent transport, gravitational settling and Brownian transport were well-represented in this study. 

## 5. Conclusions

This study established a novel method for evaluating the effects of air purification interventions on the dosage delivered to the small airway. An in vitro airway replica was developed by 3D-printing a reconstruction of CT images of a 34-year-old healthy male subject. A breathing airflow simulator was connected to the replica to mimic periodic inhalation. 

Although both had similar CADRs, the NIAP diluted airborne particles slightly faster than the filtration purifier in the test room. Although the potential inhalation doses were similar between the purification methods, the NIAP reduced the dose delivered to the small airway by up to 20% compared to the filtration purifier due to particle coagulation, even though the NIAP shifted the particle size distribution by less than 5% for all size bins. The coagulation effect of the NIAP was strongest on particles less than 0.686 µm. Higher negative ion concentrations lead to lower dosages delivered to both the breathing zone and the small airway. 

To conclude, NIAPs represent an energy-efficient air purification intervention that can effectively reduce particle exposure to the small airway, provided sufficient negative ion concentration is maintained around the subject. Further studies on the health consequences of NIAPs that assess detailed and systematic molecular mechanisms will enhance the search for air purification interventions and meet both health and energy demands.

## Figures and Tables

**Figure 1 ijerph-19-00264-f001:**
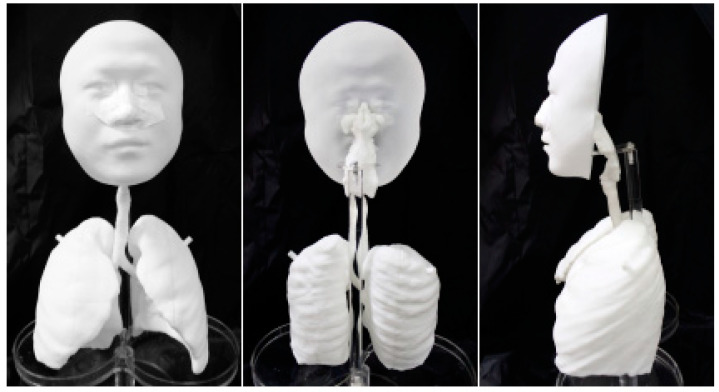
The in vitro human airway replica. The face and upper airway include the nasal and oral cavities and the lower airway includes five generations of bronchi and two realistic lung cavities. The nose was blocked during measurement to simplify the oral inhalation exposure. The breathing simulator was connected to the conducting tubes below the two lung cavities to mimic periodic inhalation.

**Figure 2 ijerph-19-00264-f002:**
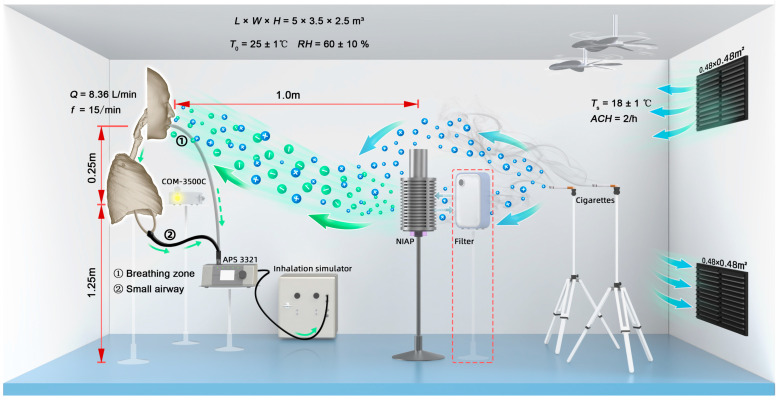
Experimental set-up of the full-scale exposure measurement. Momentum and turbulent transport of particles dominated the inhalation exposure mechanism in the well-mixed condition. Transport via entrainment of the body thermal boundary layer was omitted.

**Figure 3 ijerph-19-00264-f003:**
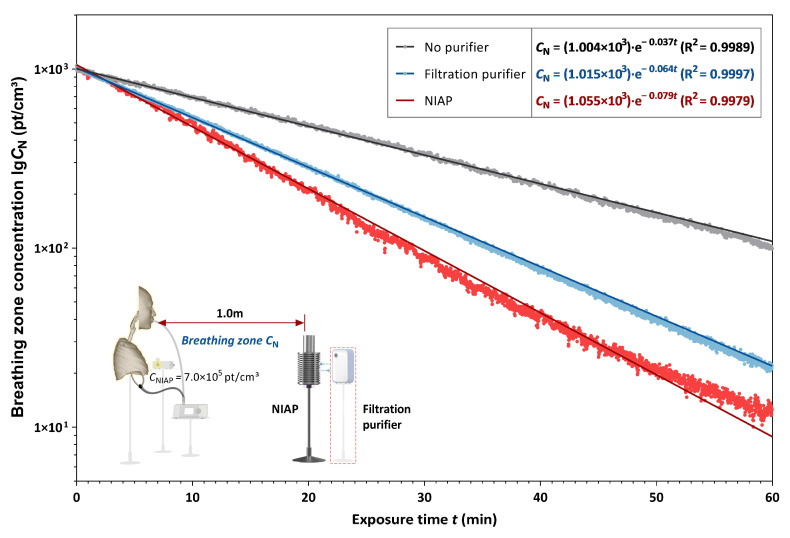
The PM_10_ concentration decay in the breathing zone during the 1 h exposure. Gray dots show the natural decay (no purifier); blue dots show the decay with the filtration purifier; red dots show the decay with coagulation by the NIAP. The dilution coefficient of NIAP was the largest at 0.079, followed by 0.064 for the filtration purifier and 0.037 for natural decay.

**Figure 4 ijerph-19-00264-f004:**
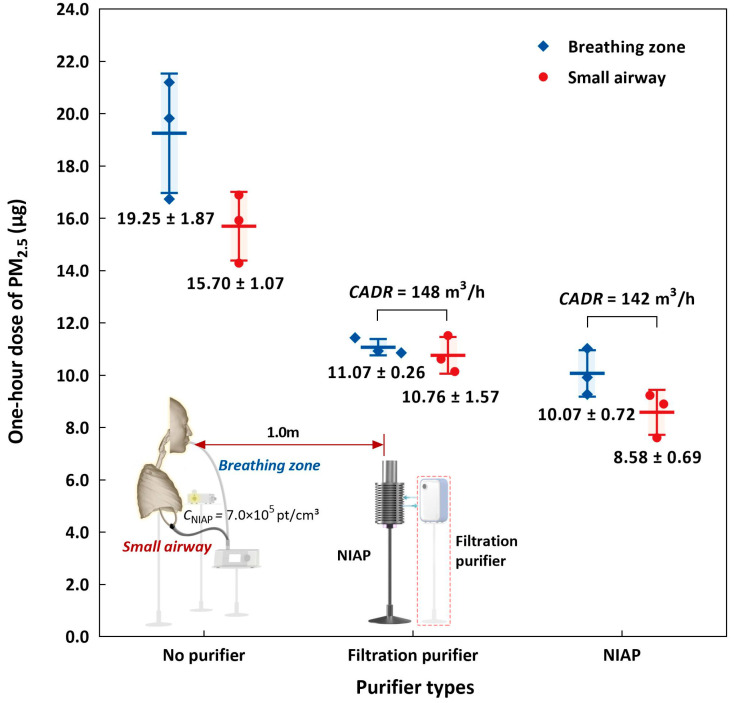
Comparison of air purification interventions on the potential inhalation dose of PM_2.5_ (in blue) and the administered dose of PM_2.5_ to the small airway (in red) after the 1 h exposure.

**Figure 5 ijerph-19-00264-f005:**
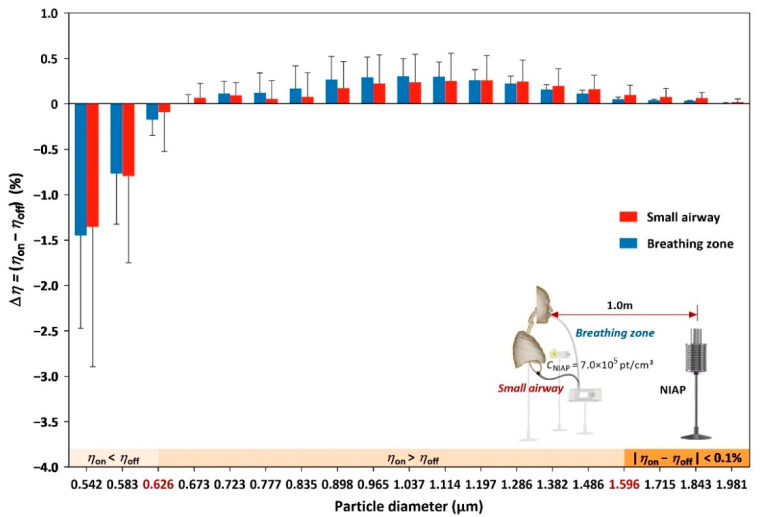
Coagulation-induced variation (*η*) in the particle number ratio in each size bin from 0.542 to 1.981 µm. *η* = *C_D_*/*C_Total_* × 100%, where C_D_ is the particle concentration in a specific size bin; *C_Total_* is the total particle concentration of all size bins; *η_on_* is the particle number ratio when the NIAP is turned on; *η_off_* refers to the particle number ratio when the NIAP is turned off. Bars in blue and red show the particle number ratio variation in the breathing zone and small airway, respectively.

**Figure 6 ijerph-19-00264-f006:**
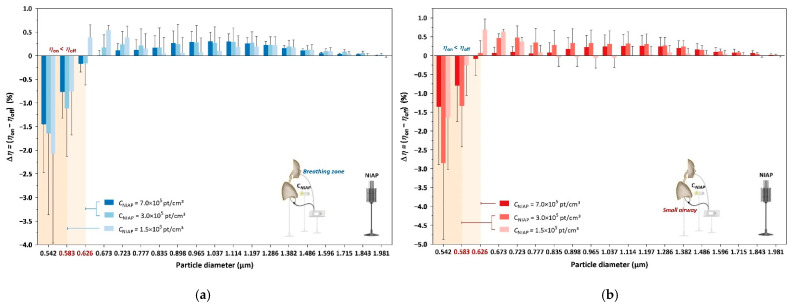
Coagulation-induced variation in the particle number ratio under negative ion concentrations (*C_NIAP_*) of 7.0 × 10^5^, 3.0 × 10^5^, 1.5 × 10^5^ pt/cm^3^ in the breathing zone (**a**) and the small airway (**b**).

**Figure 7 ijerph-19-00264-f007:**
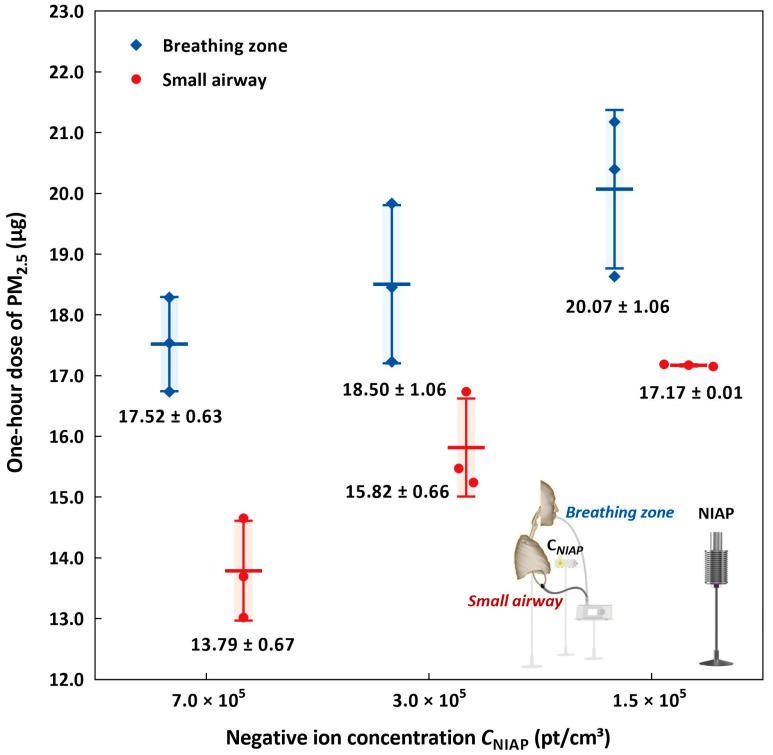
Comparison between the negative ion concentration and PM_2.5_ potential inhalation and administered dose in the small airway after the 1 h exposure.

**Table 1 ijerph-19-00264-t001:** Experimental parameters.

Index	Value
Supply air temperature *T*_s_ (°C)	18 ± 1
Air change rate ACH (per hour)	2
Cleanroom temperature *T*_0_ (°C)	25 ± 1
Cleanroom relative humidity RH (%)	60 ± 10

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
