# Peer review of "Negative Ion Purifier Effects on Indoor Particulate Dosage to Small Airways"

_ijerph, 2021, doi:10.3390/ijerph19010264_

Round 1

Reviewer 1 Report

- I think it is better to add whether the subject of the 3d model has no airway disease and is a non-smoker.

- YSM50M-0713-S-L(air-conditioning system) part is hard to find information. Please describe it in more detail or reference.

- line 141 : APS 3321 (TSI) : Please add device manufacturers, regions, and countries.

- line 141 : Please add the range of particle size distributions measured by the device.

- line 164: Why did the author measure PM10? Cigaret smoke is mainly ultrafine particles below pm1.0.

Author Response

Thanks for the suggestions. We have revised our manuscript accordingly. The manuscript has also been carefully revised by a native English editor to correct all grammatical errors.

Dear Reviewer 1:

Thanks for reviewing our manuscript and giving us constructive comments. The manuscript has been carefully revised. Please find our response to each specific comment below, and the brief description of the revision as highlighted in blue fonts. The revised portion is marked in red fonts of the revision in the manuscript.

Comments to the Author

  1. I think it is better to add whether the subject of the 3d model has no airway disease and is a non-smoker.

Thanks for the suggestion. The subject is a non-smoker with no airway disease. We added it to Line 113 on Page 3.

  1. YSM50M-0713-S-L (air-conditioning system) part is hard to find information. Please describe it in more detail or reference.

Thanks for the suggestion. “An air-conditioning system YSM50M-0713-S-L (YORK Incorporated, China) in the room could control the indoor temperature and humidity. YSM50M-0713-S-L (380V, 3N, 50Hz) has a rated ventilation rate of 4500 m3/h, with a cooling capacity of 30 kW and a heating capacity of 18 kW.”  We added this description to Line 124 on Page 3.

  1. Line 141: APS 3321 (TSI) : Please add device manufacturers, regions, and countries.

Thanks for the suggestion. The Aerodynamic Particle Sizer Spectrometer Model APS 3321 is manufactured by TSI Incorporated, Minnesota, USA. We added this description to Line 141 on Page 4.

  1. Line 141 : Please add the range of particle size distributions measured by the device.

Thanks for the suggestion. APS 3321 provides count size distributions for particles with aerodynamic diameters from 0.523 to 19.810 µm. We added this description to Line 4 on Page 144.

  1. Line 164: Why did the author measure PM10? Cigarette smoke is mainly ultrafine particles below pm1.0.

Thanks for the question. We adopted this specific cigarette smoke, because it is a recommended standard particle source. It is correct that most of particles have an aerodynamic diameter below 1.0 µm. However, we still measured about 5% particles in the range between 1.114 to 2.839 µm, as well as <1% between 2.839 and 8.354 µm. Therefore, we compared the decay of total PM10 instead of PM2.5 or PM1.0, when we compared particle concentration decay among natural deposition, NIAP and filtration.

Reviewer 2 Report

The paper titled, ‘Assessing the Effect of Negative Ion Purifier on Indoor Particulate Dosage to Small Airway’ by Duan and colleagues focuses on the construction of an in-vitro airway model and the contribution of negative ion purifiers is quantified. This paper merits publication after the revisions are made to the original manuscript based on the below mentioned comments.

Line 15: The authors need to add the word ‘time’……as we spend more than 80% time indoors…..

Line 38: Revise the sentence as …’does a specific link….’does not make sense grammatically.

Line 40: How about other indoor sources of pollutants such as pet dander, allergens, volatile organic compounds..?

Line 45: I think the authors are missing a word after ‘clinical’.

Line 48: You need to provide references here.

Line 51-53: Please rewrite the sentence again as the sentence structure is confusing. It would be better if the manuscript is revised with the help of a native English speaker or writer.

Line 53: Why just focus on the last century ( i.e. 20th century)? What about the twenty one years of this century.

Line 55: Change ‘limit’ to limited because limit is a verb. I think the authors mean limited number of subjects.

Line 57: What is the controversy the authors are alluding to? Controversy is a very loaded word. Please elaborate else remove it.

Line 58: Spelling error: Purififaction???? Do the authors mean purification?

Line 58 again: Spelling error – performace…..

Line 64: Change ‘ declared’ to ‘indicate’ or ‘suggest’

Line 74: Please remove the semi-colon.

Line 77: Change ‘ seem to be getting another voice’….to ……’two recent NIAP interventions studies have yielded different results’….

Line 128: Change measurement to ‘measurements’

Line 181: I think the authors means….better purification ability for PM10 and not …of PM10.

Line 182-196: Ok, this is very funny. It seems the authors have copy-pasted the comments of a previous reviewer when they had submitted it to another journal. I am not sure why the authors included this here !

Line 224: Instead of starting the sentence with ‘While’ please start as follows:….In contrast, NIAP reduced the dose……’

Line 277: Use ‘that’ before requires…. ….NIAP is a method ‘that’ requires less than 1/6 energy consumption of filtration…

Line 293: Please remove this line…it sounds very theatrical.

Author Response

Thanks for the suggestions. We have revised our manuscript accordingly. The manuscript has also been carefully revised by a native English editor to correct all grammatical errors.

Dear Reviewer 2:

Thanks for reviewing our manuscript and giving us constructive comments. The manuscript has been carefully revised. Please find our response to each specific comment below, and the brief description of the revision as highlighted in blue fonts. The revised portion is marked in red fonts of the revision in the manuscript.

Comments to the Author

The paper titled, ‘Assessing the Effect of Negative Ion Purifier on Indoor Particulate Dosage to Small Airway’ by Duan and colleagues focuses on the construction of an in-vitro airway model and the contribution of negative ion purifiers is quantified. This paper merits publication after the revisions are made to the original manuscript based on the below mentioned comments.

Thanks for the suggestion. The language expression of our manuscript has been revised by the professional language editing. We list our response to each question below.

Grammatical suggestions

  1. Line 15: The authors need to add the word ‘time’……as we spend more than 80% time indoors…..
  2. Line 38: Revise the sentence as …’does a specific link….’does not make sense grammatically.
  3. Line 45: I think the authors are missing a word after ‘clinical’.
  4. Line 51-53: Please rewrite the sentence again as the sentence structure is confusing. It would be better if the manuscript is revised with the help of a native English speaker or writer.
  5. Line 55: Change ‘limit’ to limited because limit is a verb. I think the authors mean limited number of subjects.
  6. Line 58: Spelling error: Purififaction???? Do the authors mean purification?
  7. Line 58 again: Spelling error – performace…..
  8. Line 64: Change ‘ declared’ to ‘indicate’ or ‘suggest’
  9. Line 74: Please remove the semi-colon.
  10. Line 77: Change ‘ seem to be getting another voice’….to ……’two recent NIAP interventions studies have yielded different results’….
  11. Line 128: Change measurement to ‘measurements’
  12. Line 181: I think the authors means….better purification ability for PM10 and not …of PM10.
  13. Line 224: Instead of starting the sentence with ‘While’ please start as follows:….In contrast, NIAP reduced the dose……’
  14. Line 277: Use ‘that’ before requires…. ….NIAP is a method ‘that’ requires less than 1/6 energy consumption of filtration…

Thanks for the careful proofing. Above grammatical errors have been corrected and revised based on suggestions from a professional language editor.

  1. Line 40: How about other indoor sources of pollutants such as pet dander, allergens, volatile organic compounds..?

Thanks for the suggestion. We elaborate the claim into a new sentence, ‘Indoor PM is a complex mixture with varied compositions, origins and adverse health effects; the efficient control of in-door particulate exposure can substantially improve public health.” 

  1. Line 48: You need to provide references here.

Thanks for the suggestion. We added a reference to “Grinshpun, S.A., Mainelis, G., Trunov, M., Adhikari, A., Reponen, T. and Willeke, K. (2005), Evaluation of ionic air purifiers for reducing aerosol exposure in confined indoor spaces. Indoor Air, 15: 235-245. https://doi.org/10.1111/j.1600-0668.2005.00364.x”

  1. Line 53: Why just focus on the last century ( i.e. 20th century)? What about the twenty one years of this century.

Thanks for the question. We were trying to express last one hundred years before present days, instead of 20th century. Sorry for this misleading error. We have changed the expression to “Numerous studies have discussed the health consequences of exposure to airborne ions and reported conflicting conclusions”.

  1. Line 57: What is the controversy the authors are alluding to? Controversy is a very loaded word. Please elaborate else remove it.

Thanks for the question. We didn’t elaborate the controversy before this sentence, since we cited a systematical review study by Alexander et al (2013), Reference 14. We now added two sentences before this claim. “Some studies have found that exposure to negative air ions has a beneficial effect on respiratory health [14–16]. In contrast, other studies have reported that exposure to ions adversely affects lung function and cardiovascular performance, causes irritation and potentially exacerbates asthma symptoms [16–20]. Unfortunately, due to limited sample sizes and/or significant methodology biases in these studies, the health effects of NIAPs remain unclear and controversial [21].” 

  1. Line 182-196: Ok, this is very funny. It seems the authors have copy-pasted the comments of a previous reviewer when they had submitted it to another journal. I am not sure why the authors included this here !

We apologize for the mistake. This manuscript has not been submitted anywhere else or before. These three paragraphs are the original text in the submission template of IJERPH, and we forgot to remove them. 

  1. Line 293: Please remove this line…it sounds very theatrical.

Thanks for the question. It has been removed.

Reviewer 3 Report

Review: Assessing the Effect of Negative Ion Purifier on Indoor Particulate Dosage to Small Airway

This paper confirms the contribution of negative ion air purifier (NIAP) to 0.5-2.5 um particles in a model produced by CT scan and 3D printer technology of indoor particulates for human airways.

The experiment was carried out in a clean room using a 3D model based on a CT scan of a healthy 34-year-old male, and the negative ion concentration and microparticles were measured by dispersing the micro particles.

As a result of the measurement, it was confirmed that the higher the concentration of the negative ion, the more than about 20% of the exposure to fine particles could be reduced compared to the general filter purifier.

This paper efficiently identified the effect of NIAP on microparticles and airways. Accordingly, this paper is sufficient for publication in the international Journal of Environmental Research and Public Health.

The detailed reasons are as follows.

- The 3D human body model was made to resemble a human airway.

- The test environment was well set, and the size of the micro particles and the concentration of negative ions were set constant.

- The micro particulate exposure formula was appropriately calculated and the blank was also measured to increase the accuracy of the measurement.

- The results were well compared with NIAP and CADR, and conclusions were reached.

- The effect of anion concentration was explained in detail.

- Appropriate opinions of the author are included.

- We made clear conclusions about the coagulation effect of NIAP.

Author Response

Thanks for your kind support.